# Liquid Biopsy Targeting Monocarboxylate Transporter 1 on the Surface Membrane of Tumor-Derived Extracellular Vesicles from Synovial Sarcoma

**DOI:** 10.3390/cancers13081823

**Published:** 2021-04-11

**Authors:** Suguru Yokoo, Tomohiro Fujiwara, Aki Yoshida, Koji Uotani, Takuya Morita, Masahiro Kiyono, Joe Hasei, Eiji Nakata, Toshiyuki Kunisada, Shintaro Iwata, Tsukasa Yonemoto, Koji Ueda, Toshifumi Ozaki

**Affiliations:** 1Department of Orthopaedic Surgery, Okayama University Graduate School of Medicine, Dentistry, and Pharmaceutical Sciences, 2-5-1, Shikata-cho, Kita-ku, Okayama 700-8558, Japan; d04sm100@yahoo.co.jp (S.Y.); akysda@gmail.com (A.Y.); m.takuyasungoi@gmail.com (T.M.); asuka02820282@yahoo.co.jp (M.K.); joe@md.okayama-u.ac.jp (J.H.); eijinakata8522@yahoo.co.jp (E.N.); toshikunisada@gmail.com (T.K.); tozaki@md.okayama-u.ac.jp (T.O.); 2Department of Orthopaedic Surgery, Okayama Rosai Hospital, 1-10-25, Chikkomidorimachi, Minami-ku, Okayama 702-8055, Japan; coji.uo@gmail.com; 3Department of Orthopaedic Surgery, Chiba Cancer Center, 666-2, Nitona-cho, Chuo-ku, Chiba 260-8717, Japan; shiwata@ncc.go.jp (S.I.); tyonemot@chiba-cc.jp (T.Y.); 4Cancer Precision Medicine Center, Japanese Foundation for Cancer Research, 3-8-31, Ariake, Koto, Tokyo 135-8550, Japan; koji.ueda@jfcr.or.jp

**Keywords:** liquid biopsy, synovial sarcoma, monocarboxylate transporter 1, extracellular vesicles, non-invasive biomarker

## Abstract

**Simple Summary:**

Synovial sarcoma (SS) is associated with a high risk of recurrence and poor prognosis, and no biomarker useful in monitoring tumor burden exists. We identified monocarboxylate transporter 1 (MCT1) expressed in extracellular vesicles (EVs) derived from synovial sarcoma as a potential such marker. Circulating levels of MCT1^+^CD9^+^ EVs were significantly correlated with tumor volume in a SS mouse model. Serum levels of MCT1^+^CD9^+^ EVs reflected tumor burden and treatment response in SS patients. Patients with MCT1 expression on the plasma membrane have significantly worse overall survival than those with nuclear expression. Silencing of MCT1 reduced the malignant phenotype including cellular viability, migration, and invasion of SS cells. MCT1 may thus be a promising novel target for liquid biopsies and a novel therapeutic target.

**Abstract:**

The lack of noninvasive biomarkers that can be used for tumor monitoring is a major problem for soft-tissue sarcomas. Here we describe a sensitive analytical technique for tumor monitoring by detecting circulating extracellular vesicles (EVs) of patients with synovial sarcoma (SS). The proteomic analysis of purified EVs from SYO-1, HS-SY-II, and YaFuSS identified 199 common proteins. DAVID GO analysis identified monocarboxylate transporter 1 (MCT1) as a surface marker of SS-derived EVs, which was also highly expressed in SS patient-derived EVs compared with healthy individuals. MCT1^+^CD9^+^ EVs were also detected from SS-bearing mice and their expression levels were significantly correlated with tumor volume (*p* = 0.003). Furthermore, serum levels of MCT1^+^CD9^+^ EVs reflected tumor burden in SS patients. Immunohistochemistry revealed that MCT1 was positive in 96.7% of SS specimens and its expression on the cytoplasm/plasma membrane was significantly associated with worse overall survival (*p* = 0.002). Silencing of MCT1 reduced the cellular viability, and migration and invasion capability of SS cells. This work describes a new liquid biopsy technique to sensitively monitor SS using circulating MCT1^+^CD9^+^ EVs and indicates the therapeutic potential of MCT1 in SS.

## 1. Introduction

Soft-tissue sarcomas (STSs) are a heterogeneous group of malignant tumors of mesenchymal origin with more than 50 histological subtypes [1,2]. STSs constitute a particularly rare group of cancers, comprising less than 1% of all malignancies [3,4,5]. While surgical resection remains the mainstay of treatment, multidisciplinary treatments, including chemotherapy and/or radiotherapy provide a survival benefit for patients with STS that is sensitive to these therapies [6,7,8,9,10]. Despite the advancement in multidisciplinary treatments, survival outcomes have almost plateaued for the past 10 years. One of the reasons for this is the lack of available biomarkers to monitor the tumor response to multimodal treatment or tumor relapse following definitive treatment. Synovial sarcoma (SS), a high-grade tumor accounting for 6–10% of all STSs, is no exception and represents a high risk of tumor relapse, estimated to be 12% locally and 39% at distant sites [11]. SS is characterized by the specific chromosomal translocation t(X;18) (p11.2;q11.2), which leads to the chimeric fusion gene *SS18-SSX* [12]. While the presence of chromosomal translocation in tumor specimens has been used for clinical diagnosis, no biomarker that is capable of monitoring tumor burden or treatment response has yet been identified.

Emerging evidence has suggested that liquid biopsy as a non-invasive method has a promising future in clinical oncology, and the field of extracellular vesicles (EVs) has drawn considerable attention recently. EVs are small membranous vesicles composed of a lipid bilayer with a cystic structure; EVs are naturally secreted by almost all cell types and can transport bioactive molecules intercellularly [13,14]. Studies have shown that tumor-released EVs are found in the blood of patients with cancer [15,16,17,18], indicating a novel use as a noninvasive biomarker for patients with various types of malignancy. Since EVs express tetraspanin family proteins such as CD63, CD81, and CD9 [19,20], circulating EVs could be monitored by the detection of these proteins and other markers specific to tumor-specific EVs. A novel liquid biopsy technique using ExoScreen, a sensitive and rapid detection method for profiling circulating exosomes directly from blood samples, was shown to be able to monitor circulating EVs using the antigens CD147 and CD9 in patients with colorectal cancer [14]. However, the surface markers of EVs secreted from sarcoma cells remain to be elucidated and, thus, no technology has been established for tumor monitoring by the detection of EVs secreted from sarcomas. In this study, we identified surface markers of EVs secreted from SS cells, in order to establish a liquid biopsy technique that can be used to monitor the tumor burden or response to treatments for SS.

## 2. Materials and Methods

### 2.1. Cell Culture

The human SS cell lines Aska-SS, HS-SY-II, SYO-1, YaFuSS, and Yamato-SS were used in this study. SYO-1 was previously established in our laboratory [21], and Aska-SS, HS-SY-II, YaFuSS, and Yamato-SS were kindly provided by Toguchida, Sonobe, and Naka, respectively [22,23]. Cell lines were cultured in Dulbecco’s modified Eagle’s medium (DMEM; Gibco Laboratories, Grand Island, NY, USA) supplemented with 10% fetal bovine serum (FBS; Hyclone Laboratories Inc., Logan, UT, USA), 100 units/mL of penicillin G (NACALAI TESQUE, Inc., Kyoto, Japan). Cells were incubated at 37 °C in a humidified atmosphere containing 5% CO_2_.

### 2.2. Preparation of Conditioned Medium

The conditioned medium (CM) was changed to FBS-free CM at 24 h after seeding of cells, and then collected at 24 h after CM exchange. The collected CM was centrifuged at 3500 rpm for 15 min at 4 °C. The CM supernatant was collected and centrifuged at 20,000× *g* for 15 min at 4 °C, and supernatants were collected and passed through a 0.22-μm-pore filter (Merck Millipore, Billerica, MA, USA) before storage at −80 °C.

### 2.3. Patient Serum and Tissue Samples

All experimental methods were carried out in accordance with relevant guidelines and regulations. Whole blood samples were obtained from 17 patients with SS at the time of diagnosis and four healthy individuals. Seven of the samples, collected from 3 patients with SS and 4 healthy individuals, were used for the proteomic analysis. Tumor monitoring was performed using EVs in 10 patients; the blood samples used were obtained at the time of diagnosis, before chemotherapy, after chemotherapy, or after surgery, or at disease progression stage. Murine blood was obtained by cardiac puncture at the indicated time points. Serum was fractionated from whole blood samples by centrifugation at 3500 rpm for 15 min at 4 °C. The collected serum was centrifuged at 20,000× *g* for 15 min at 4 °C, and supernatants were collected and passed through a 0.22-μm-pore filter (Merck Millipore) before storage at stored −80 °C. Tumor samples from patients with SS were collected between 1969 and 2015 at the Okayama University Hospital.

### 2.4. Isolation of EVs from Cell CM

SS cells were grown to 50–70% confluence and then CM was exchanged to FBS-free CM. The CM was collected 24 h after medium exchange and was centrifuged at 3500 rpm for 15 min at 4 °C, followed by further centrifugation at 9000× *g* for 30 min at 4 °C. The supernatant was then passed through a 0.22-μm-pore filter (Merck Millipore) to remove apoptotic bodies and cell debris. The collected CM supernatant was concentrated to approximately 1 mL using 100 kDa MWCO ultrafiltration membranes (Fisher Scientific, Loughborough, UK) at 4 °C. The sample was then ultracentrifuged using Optima TL-100 (Beckman Coulter, Fullerton, CA, USA) at 100,000× *g* for 70 min at 4 °C. The supernatant was discarded, and the EV pellet was rinsed with PBS, followed by further ultracentrifugation at 100,000× *g* for 70 min at 4 °C. Finally, the supernatant was discarded and the EVs were concentrated in the pellet. The obtained EVs were authenticated by transmission electron microscopy (TEM), and the diameters of the particles present in the EV fractions were analyzed by a Zetasizer nano ZSP (Malvern Panalytical, Malvern, UK).

### 2.5. Isolation of EVs from Human Serum

EVs were purified from human serum samples using size exclusion chromatography on a drip with extracellular vesicle isolation by size exclusion chromatography on a EVSecond L70^®^ drip column (GL Sciences, Tokyo, Japan). The column was initially equilibrated with 700 μL of FBS twice, followed by three washing steps using 1500 μL of PBS. After washing, 100 μL of the collected human serum sample was loaded onto the column, followed by the collection of 12 consecutive fractions in 100 μL of PBS. The CD9 expression in these fractions was analyzed using western blotting, and CD9-positive fractions were recognized as the exosome-rich portion. The obtained EVs using this system were previously authenticated with TEM and western blotting, and the diameters of the particles in the EV fractions were analyzed with the Zetasizer nano ZSP [24].

### 2.6. Liquid Chromatography-Tandem Mass Spectrometry (LC/MS) Analysis

The EV-containing eluates of EVSecond L70 columns (GL Sciences, Tokyo, Japan) were dried and resolved in 20 mM HEPES-NaOH (pH 8.0), 12 mM sodium deoxycholate, and 12 mM sodium *N*-lauroyl sarcosinate. Following reduction with 20 mM dithiothreitol (DTT), at 100 °C for 10 min and alkylation with 50 mM iodoacetamide at ambient temperature for 45 min, proteins were digested with 5 μL of immobilized trypsin (Thermo Fisher Scientific, Inc., Waltham, MA, USA) with shaking at 1000 rpm at 37 °C for 6 h. After the removal of sodium deoxycholate and sodium *N*-lauroyl sarcosinate by ethyl acetate extraction, the resulting peptides were desalted by Oasis HLB μElution plate (Waters Corp., Milford, MA, USA) and subjected to mass spectrometric analysis. Peptides were analyzed using an LTQ-Orbitrap-Velos mass spectrometer (Thermo Fisher Scientific, Inc.) combined with an UltiMateTM 3000 RSLCnano-flow HPLC system (Thermo Fisher Scientific, Inc.). Protein identification and quantification were performed using MaxQuant software (https://www.biochem.mpg.de/5111795/maxquant). The MS/MS spectra were searched against the Homo sapiens protein database in SwissProt, with a false discovery rate set to 1% for both peptide and protein identification filters. Only “Razor + unique peptides” were used for the calculation of relative protein concentration. Candidate proteins were selected using the following criteria: commonly expressed in EVs from all of the SS cell lines; plasma membrane protein expressed on the EV membrane; higher expression level in patients with SS than in healthy individuals; and decreased expression level postoperatively in patients with SS.

### 2.7. Transmission Electron Microscopy (TEM)

The purified EVs were measured in TEM images according to the method described by Eguchi et al. [25]. Briefly, a 400-mesh copper grid coated with formvar/carbon films was hydrophilically treated [25]. The EV suspension (5–10 μL) was placed on parafilm, and the grid was floated on the EV liquid and left for 15 min. The sample was negatively stained with 2% uranyl acetate solution for 2 min. EVs on the grid were visualized at 20,000× magnification with an H-7650 transmission electron microscope (Hitachi, Tokyo, Japan) at the Central Research Laboratory of Okayama University Medical School.

### 2.8. Particle Diameter Analysis

The diameters of the EVs (40 μL) were analyzed using the Zetasizer nano ZSP (Malvern Panalytical, Malvern, UK).

### 2.9. Animal Experiments

BALB/c nu/nu female mice were purchased at 4 weeks of age (CLEA Japan Inc., Tokyo, Japan). Yamato-SS cells (1 × 10^7^ cells in 100 μL of PBS) were inoculated subcutaneously on the left buttock under general anesthesia performed with 2% isoflurane. Tumor sizes were measured after 1 week of injection and subsequently measured once a week. One of three groups (*n* = 9; tumor bearing/resection group) had tumor resection 5 weeks after tumor inoculation, followed by blood collection 1 week after tumor resection to avoid the bias by the post-surgical inflammation. Other groups (tumor-bearing/no resection group, and non-tumor bearing group) had blood collection by cardiac puncture 5 weeks after injection under general anesthesia. Animals were euthanized after blood collection.

### 2.10. Immunohistochemistry

The data of 30 patients with SS were analyzed. Slides were heated for antigen retrieval in 10 mM sodium citrate (pH 6.0), followed by incubation with 1:200 anti-MCT1 mouse monoclonal antibody (sc-365501; Santa Cruz Biotechnology, Santa Cruz, CA, USA) or isotype-matched control antibodies overnight at 4 °C. Immunodetection was performed using Histofine anti-mouse IgG (Nichirei, Tokyo, Japan) and the DAB substrate kit (Nichirei) according to the manufacturer’s instructions. Sections were counterstained with hematoxylin to create contrast. Sections were scored semi-quantitatively for cytoplasmic expression as follows: score 0, 0% immunoreactive cells; score 1, <5% immunoreactive cells; score 2, 5–50% immunoreactive cells; score 3, >50% immunoreactive cells. The intensity of staining was scored semi-qualitatively as follows: 0, negative; 1, weak; 2, intermediate; and 3, strong. The final score was defined as the sum of both parameters (extension and intensity) and grouped as negative (score 0 and 2) and positive (score 3–6), as previously described [26].

### 2.11. Immunoblot Analysis

Total protein from cells (10 μg) or EVs (30 μg) was fractionated using an electrophoretic gradient across Mini-PROTEAN^®^ tris-glycine extended gels (BIO-RAD, Richmond, CA, USA). Loading samples were normalized according to the protein concentrations quantified using the Bradford assay. The gels were then transferred onto the Immun-Blot^®^ PVDF membrane (BIO-RAD) under wet electrophoretic conditions. The blotted protein was blocked for 1 h at room temperature with Odyssey^®^ blocking buffer in PBS (LI-COR, Lincoln, NE, USA) followed by incubation overnight at 4 °C with the following primary antibodies: 1:200 anti-MCT1 mouse monoclonal antibody (sc-365501; Santa Cruz Biotechnology); 1:200 anti-CD81 mouse monoclonal antibody (sc-23962; Santa Cruz Biotechnology); 1:2000 anti-tubulin hFAB Rhodamine (BIO-RAD). Thereafter, IRDye^®^ 800CW anti-mouse IgG secondary antibodies (LI-COR) were incubated with the protein-blotted membrane for 45 min at room temperature. Fluorescence was detected using the Odyssey^®^ imaging system (LI-COR).

### 2.12. Enzyme-Linked Immunosorbent Assay

The 250 ng/well of anti-CD9 antibody was immobilized to a Nunc MaxiSorp flat-bottom 96-well plate (Thermo Fisher Scientific). Blocking solution (150 μL/well of 5% BSA in PBS) was then added before incubation on a plate shaker at ambient temperature for 60 min. After three washes with PBS, 30 μL EVs + 70 μL PBS was loaded. Following a 2 h incubation, plates were washed three times with PBS. The 100 μL/well of biotinylated anti-CD9 antibody (500 ng/mL) or biotinylated anti-MCT1 antibody (5 μg/mL, Millipore, Cambridge, UK) in 1% BSA was loaded to the wells. After 60 min incubation, plates were washed three times with PBS and then covered with 100 mL/well of 1 × HRP-Streptavidin (Abcam) in 1% BSA. After 45 min incubation, the plates were washed three times with PBS and covered with 100 mL/well of TMB Ready Solution (Thermo Fisher Scientific). The reaction was stopped after 15 min incubation using 100 mL/well of 2N HCl. The optical density (OD) at 450 nm was immediately measured.

### 2.13. RNA Extraction and RT-qPCR

Total RNA was extracted using an miRNeasy mini Kit (Qiagen, Valencia, CA, USA) in accordance with the manufacturer’s instructions. RNA samples were reverse transcribed using the TaqMan MicroRNA Reverse Transcription Kit (Applied Biosystems, Foster City, CA, USA). For mRNA detection, total RNA was reverse transcribed using a PrimeScript RT Reagent Kit (Takara, Tokyo, Japan) in accordance with the manufacturer’s protocol. Quantitative polymerase chain reaction (PCR) was performed on an Agilent Mx3000P QPCR System (Agilent Technologies, Santa Clara, CA, USA) using the TaqMan 2 × Universal PCR Master mix and each primer. Data obtained from RT-PCR were analyzed using the 2−∆∆Ct method. The mRNA expression levels were normalized using GAPDH.

2.14. siRNA Transfection

Cells were seeded into 6- or 24-well plates on the day before the experiments. Synthetic siRNAs for MCT1 (si-MCT1, Thermo Fisher Scientific) were added to the cells at a final concentration of 20 nM in complete media the following day. DharmaFECT1 (Horizon Discovery Ltd., Cambridge, UK) were used for the transfection.

#### 2.14.1. Cell Proliferation Assay

Cellular proliferation was measured using the WST-1 Proliferation Assay (Sigma-Aldrich, St. Louis, MO, USA) in accordance with the manufacturer’s instructions. Cells were seeded in 96-well plates after 24 h of transfection. Absorbance was measured at 450 nm, with a reference wavelength of 630 nm, using a microplate reader (BIO-RAD). The relative number of viable cells is expressed as the percentage of viable cells.

#### 2.14.2. Wound Healing Assay

A scratch wound-healing assay was used to examine the cellular mobility characteristics. Cells were seeded in 24-well plates after 24 h of transfection. The medium was replaced with serum-free DMEM. Cell monolayers were scratched (wounded) using a sterile 200-μL pipette tip, and PBS was used for washing and removing cell debris. After 16 h, migrating cells were monitored and photographed under phase-contrast microscopy. The Image J software was used to quantify the relative wound size. Cell mobility inhibition (%) was calculated as the new scratch width/original scratch width × 100%. Experiments were repeated three times.

#### 2.14.3. Transwell Assay

Cell invasion and migration were examined using 24-well BD BioCoat invasion chambers with and without a Matrigel matrix (BD Biosciences, San Jose, CA, USA). SS cells (Aska-SS and Yamato-SS) were trypsinized and seeded (5 × 10^4^ cells) in chambers 48 h after siRNA transfection. The cells were subsequently suspended in FBS-free medium and added onto the upper chamber, while medium containing 10% FBS was placed in the lower chamber. After a 36 h of incubation, the cells on the upper chamber were removed, and the filters were fixed in methanol and stained with Hemacolor^®^ solution 3 (Merck, Darmstadt, Germany). The number of cells was counted in six separated high-power fields.

### 2.15. Statistical Analysis


Differences in patient demographics and clinical characteristics were measured using the chi-square test or unpaired *t*-test. Results were depicted as mean ± standard deviation or median with a 25–75% range. The Kaplan–Meier method and log-rank test were used to compare patient survival. Overall survival was defined as the period from the date of diagnosis to the censored date of death by any cause or last follow-up for survivors. Differences in MCT1^+^CD9^+^ and CD9^+^CD9^+^ levels were determined using an unpaired *t*-test or one-way analysis of variance followed by the Holm–Sidak multiple comparisons test. A ROC curve analysis was performed to examine the diagnostic potential of serum MCT1^+^CD9^+^ and CD9^+^CD9^+^ levels. A multivariate analysis was performed with the Cox regression hazard model using the clinicopathological factors that were significantly associated with overall survival in the univariate analysis. Correlations between MCT1^+^CD9^+^ or CD9^+^CD9^+^ level and in vivo tumor size were assessed with Pearson’s correlation coefficient. Differences with *p* values < 0.05 were considered statistically significant. All statistical analyses were performed using GraphPad Prism version 7.0 (GraphPad Software, San Diego, CA, USA).


## 3. Results

### 3.1. Isolation and Visualization of EVs from SS cells

In order to confirm the secretion of EVs from SS cell lines, the EVs isolated from Aska-SS, HS-SY-II, SYO-1, YaFuSS, and Yamato-SS cell lines by ultracentrifugation were analyzed under TEM. TEM revealed that the isolated particles were essentially homogeneous vesicles with sizes ranging from 40–200 nm and with a bilayer goblet structure (Figure 1A). Dynamic light-scattering analysis using Zetasizer Nano ZSP confirmed the presence of vesicles with a single peak size ranging from 139–200 nm (Figure 1B). Western blotting of the EV fractions confirmed expression of the tetraspanin protein, CD81, which is a known EV marker (Figure 1C).

### 3.2. Enrichment of MCT1 on EVs Secreted from SS Cell Lines

EVs derived from several SS cell lines were subjected to proteomic analysis, which detected 685 proteins from SYO-1-derived EVs, 301 proteins from HS-SY-II-derived EVs, and 409 proteins from YaFuSS-derived EVs (Figure 1D); among which, 199 proteins were common to EVs from all of the SS cell lines (Figure 1D, Appendix A). We next performed DAVID GO analysis to select plasma membrane protein expressed on the EV membrane. MCT1 was identified among the candidate plasma membrane proteins, and showed higher expression in patients with SS compared to healthy individuals (Appendix A). Moreover, MCT1 was also found to decrease post-operatively in SS patients (Appendix A). Western blot analysis of SS-derived EVs confirmed the expression of MCT1 in EVs from all of the SS cell lines examined, as well as in all of the SS cell lysates (Figure 1C). Next, we evaluated the total protein content of SS-secreted EVs, including total MCT1. A specific ELISA sandwich assay revealed that Yamato-SS had a high overall protein content and high amount of MCT1^+^ EVs (Figure 1E,F). Fluorescent immunostaining for MCT1 on the surface of SS cells revealed that expression of MCT1 was predominantly cytoplasmic (Figure 1G).

### 3.3. Tumor Monitoring Using MCT1 and CD9 Double-Positive EVs in a SS Mouse Model

We next established an SS xenograft mouse model using the Yamato-SS cell line in order to evaluate the in vivo dynamics of MCT1 expression levels in SS serum EVs (Figure 2A). Primary tumors were macroscopically detectable in all mice 14 days after transplantation of Yamato-SS cells. Histological evaluation demonstrated that tumors consisted of atypical spindle cells arranged in a herringbone pattern, consistent with typical findings in SS (Figure 2B). Of note, immunohistochemistry revealed diffuse expression of MCT1 in the cytoplasm of tumor cells (Figure 2B). Blood samples were obtained 5 weeks after the transplantation of Yamato-SS cells in order to investigate the correlations between tumor growth and MCT1^+^CD9^+^ EV levels. We observed a significant correlation between tumor volume and MCT1^+^CD9^+^ expression levels on EVs (R = 0.860, *p* = 0.003; Figure 2C). Furthermore, we identified higher MCT1^+^CD9^+^ EV levels in the tumor-bearing group than in the non-tumor bearing group (Figure 2D). We next analyzed the serum MCT1^+^CD9^+^ EV levels in the collected blood samples just after tumor resection. MCT1^+^CD9^+^ EV levels were significantly decreased after tumor resection (Figure 2D). These results suggest that serum levels of MCT1^+^ EVs can reflect tumor burden in the SS mouse model.

### 3.4. Tumor Monitoring by Serum MCT1^+^CD9^+^ EVs in Patients with SS

Next, we analyzed the levels of MCT1^+^CD9^+^ EVs in the serum of patients with SS using a sandwich ELISA (Table 1). EVs were first purified from 100 μL of serum from patients with SS or healthy individuals using the EV-Second method. High purity CD9^+^CD9^+^ or MCT1^+^CD9^+^ EVs were obtained in fractions 5 to 7 (Figure 3A). In a comparison of MCT1^+^CD9^+^ EVs from patients with SS, ROC analysis revealed that MCT1^+^CD9^+^ levels were able to distinguish SS patients from pre- to post-operative status, with an AUC value of 0.84 (95% confidence interval [CI] = 0.65–1.00; *p* = 0.010) (Figure 3B). The expression ratio of MCT1^+^CD9^+^ in the post-operative state was significantly reduced compared to the pre-operative state (*p* = 0.010; Figure 3C).

To further investigate the clinical utility of MCT1^+^CD9^+^ EVs for tumor monitoring, we evaluated serum expression of the MCT1^+^CD9^+^ ratio, as well as white blood cell (WBC) counts, hemoglobin (Hb) levels, and platelet (Plt) counts, in patients with SS from whom we could obtain a series of serum samples during multimodal treatment. The expression ratio of MCT1^+^CD9^+^ was monitored based on the data obtained from the serum of patients before the initiation of any treatment. Case 1 (#1; Table 1) was a 26-year-old man with SS in his left knee (Figure 3D). Serum MCT1^+^CD9^+^ EVs slightly decreased after neoadjuvant chemotherapy and further decreased after tumor resection. Microscopic evaluation revealed that the percentage of viable tumor cells was 95%. Case 2 (#2; Table 1) was a 47-year-old woman with SS in her left thigh (Figure 3E). Similarly, serum MCT1^+^CD9^+^ EVs decreased after neoadjuvant chemotherapy and further decreased after tumor resection. The percentage of viable tumor cells was 50%. Case 3 (#3; Table 1) was a 64-year-old male with SS in his right forearm (Figure 3F). Case 4 (#10; Table 1) was an 11-year-old female with SS in her groin region (Figure 3G). In Cases 3 and 4, the serum MCT1^+^CD9^+^ EVs decreased after tumor resection. In all cases, the WBC, Hb counts, and Plt levels did not correlate with MCT/CD9 levels, suggesting that MCT1^+^CD9^+^ EVs were not secreted from hematocytes. Collectively, serum MCT1^+^CD9^+^ EVs accurately reflected the tumor burden, indicating their potential for clinical use for tumor monitoring in patients with SS.

### 3.5. Prognostic Significance of MCT1 Expression Levels in SS Tumor Specimens

We next investigated the MCT1 expression in SS tumor specimens. Thirty tumors of histologically confirmed SS were analyzed by immunohistochemistry for expression of MCT1 (Table 2). Overall, the expression of MCT1 was positive in 29 cases (96.7%); MCT1 expression was present on the plasma membrane in 5 cases (17.2%), in the cytoplasm in 7 cases (24.1%), and in the nucleus in 17 cases (58.6%) (Figure 4A). Interestingly, MCT1 expression on the plasma membrane and/or in the cytoplasm was significantly associated with SS18-SSX2 positivity (*p* = 0.003), higher likelihood of metastatic progression (*p* = 0.025), and oncologic outcomes (*p* = 0.010). Furthermore, patients with MCT1 expression on the plasma membrane and/or in the cytoplasm showed a significantly worse overall survival than those with nuclear MCT1 expression (*p* = 0.002, Figure 4B). With a mean follow-up of 100.9 months (range, 2–278 months), the 5-year overall survival rates were 33% and 75% in the patients with MCT1 expression on the plasma membrane/cytoplasm and in the nucleus, respectively (*p* = 0.002; Figure 4B). In the multivariate analysis, the MCT1 expression in the cytoplasm/membrane was independently associated with worse overall survival (cytoplasm/membrane: HR = 5.34 (95%CI, 1.20–23.89) and nuclear: HR = 1; *p* = 0.028; Table 3).

### 3.6. Silencing of MCT1 Inhibits Tumor Cell Proliferation, Migration, and Invasion of SS

We next investigated the effects of silencing MCT1 using an siRNA-induced gene knockdown system in order to understand the molecular function of MCT1 in SS cells. We confirmed that MCT1 expression was significantly higher in SS cells than in hMSCs (Figure 5A). The reduced expression levels of MCT1 in Aska-SS, HS-SY-II, SYO-1, YaFuSS, and Yamato-SS cell lines were observed following siRNA targeting of MCT1, and were especially apparent in Aska-SS and Yamato-SS cells (Figure 5B). Silencing of MCT1 in Aska-SS and Yamato-SS cells also reduced cell viability (Figure 5C). We next examined the effects of MCT1 silencing on cellular migration using a wound healing assay. After transfection of siMCT1 or negative control siRNA (siNC), Aska-SS and Yamato-SS cells were plated in 24-well plates and a scratch was made after 24 h. SS cells with MCT1 knock-down showed a significant delay in migration compared to the control cells (Figure 5D). We next performed invasion and migration assays, and found that silencing of MCT1 in Aska-SS and Yamato-SS cells significantly reduced the invasion and migration capabilities of Aska-SS and Yamato-SS cells (Figure 5E). These results suggest that MCT1 contributes to the malignant phenotype of SS cells, and silencing of MCT1 may act to inhibit tumor progression.

## 4. Discussion

Emerging evidence has demonstrated the clinical utility of circulating EVs as a diagnostic, non-invasive biomarker in patients with malignant tumors. Several studies have demonstrated their potential in various cancers, including breast [26,27,28,29], prostate [30,31], pancreatic [32], ovarian [33,34], colorectal cancers [14], and glioblastoma [35] but the evidence is lacking in bone and soft-tissue sarcomas. In this study, we first demonstrate that circulating SS-derived EVs may represent a novel target for liquid biopsy of this aggressive disease. We also identified the functional relevance of circulating EVs in the progression of SS.

MCT1 has a key role in energy transfer by establishing a lactate shuttle-system [36,37]. Tumor cells import and utilize lactate for oxidative energy production (reverse Warburg metabolism), and the presence of these reverse Warburg cells is associated with a more aggressive phenotype and worse prognosis in breast, prostate, endometrial or colorectal cancer [36,38,39,40,41,42]. Furthermore, high MCT1 expression in tumor cells is associated with reverse Warburg metabolism [43,44], and silencing of MCT1 decreases resistance to chemotherapy in pancreatic adenocarcinoma cells [45]. Moreover, inhibition of MCT1 induces the accumulation of protons in the cytoplasm, resulting in acidification of glioblastoma cells as a consequence of intracellular acidification [46]. Our results demonstrate the molecular function of MCT1 in human sarcoma, and showed that silencing of MCT1 contributes to the inhibition, cellular proliferation, migration, and invasion of SS cells, although the specific molecular mechanisms remain to be elucidated. Therefore, targeting MCT1-dependent lactate transport may represent a novel option for the treatment of malignant diseases.

In the current study, global proteomic analyses identified MCT1 as a surface marker of SS-derived EVs. In spite of the small sample size, we found that the circulating MCT1^+^CD9^+^ EVs in the serum reflected the tumor burden or treatment response in vivo, indicating the potential for the translation of this liquid biopsy into clinics for the management of SS. MCT1 was recently identified in exosomes derived from malignant glioma cells [47]. Thakur et al. identified MCT1 mainly in the membrane of exosomes, and observed higher MCT1 expression in serum-derived exosomes from a mouse model of glioma compared to those of wild-type mice [47]. These findings are consistent with our data, strongly suggesting that the detection of MCT1 on circulating serum could improve diagnosis and tumor monitoring of SS.

We identified significant correlation between MCT1 expression in SS tumors and prognosis; these results were consistent with those of patients with breast cancer. A strong MCT1 staining pattern in breast cancer cells was shown to be associated with larger tumor size, shorter progression free survival, and increased risk of recurrence [36]. In our study, MCT1 expression on the plasma membrane/cytoplasm was significantly associated with worse overall survival than in those with nuclear MCT1 expression. Therefore, the cellular localization of MCT1 expression can predict poor prognosis in SS. The nuclear expression of MCT1 in STSs was previously reported by Pinheiro et al. [37]. They identified MCT1 expression of STSs on the plasma membrane in 60.5% and nuclear expression in 32.6% of patients, and also demonstrated that patients with MCT1 expression on the plasma membrane had worse prognosis [37]. The cellular localization does not fit with the classic role of this protein as a transmembrane transporter. Although the role of MCT1 according to the cellular localization is not fully understood, Pinheiro et al. anticipate that MCT1 has an additional role in the nucleus that is unrelated to the lactate transport activity and may function as a tumor suppressor [37].

Emerging reports have shown the potential of circulating tumor cells (CTCs), nucleic acids, and EVs as novel targets for liquid biopsy, although these have not yet been translated into standard clinical use. Although CTCs can be captured from blood they are extremely rare within an overwhelming background: 1 mL of whole blood contains about 5 million mononuclear cells and only 1.43% of progressive breast cancer patients had more than 500 CTCs in 7.5 mL blood [48,49,50,51]. Moreover, many issues regarding the sampling bias of the captured cells remain unknown [52]. Furthermore, the limited ability to monitor early stage malignancies is another drawback of the clinical use of CTCs [53]. ctDNA has similar limitations for early stage cancers, although this provides high sensitivity with a stable molecule. In contrast, EVs have the potential to have higher sensitivity than ctDNA/CTCs in early stage cancer. Indeed, in this study, MCT1^+^CD9^+^ was detected in all of the serum samples obtained from patients with localized and metastatic SS, which seemed to accurately reflect the tumor burden. The advantages of this method include the fact that it is a relatively easy procedure with a low cost compared to methods targeting ctDNA such as next generation sequencing. However, the use of EVs for liquid biopsies has not been utilized in large clinical trials and carries the potential for contamination with EVs from normal tissue/cells. Despite these disadvantages, liquid biopsy targeting tumor-derived EVs is a promising method to extend the possibilities for tumor monitoring or prognostic prediction.

## 5. Conclusions

We identified MCT1 as a surface marker of SS-derived EVs. The circulating level of MCT1^+^CD9^+^ EVs in the serum reflected tumor burden in both SS-bearing mice and patients. Furthermore, plasma membrane expression of MCT1 was found to be associated with worse prognosis, while nuclear expression is associated with better prognosis. Silencing MCT1 inhibited SS cell migration and invasion and suppressed tumor growth. Thus, MCT1 could be a promising novel target for liquid biopsy and a novel therapeutic target. Further experiments are warranted to establish the diagnostic and therapeutic significance of MCT1, which may overcome the diagnostic and therapeutic limitations in the management of SS.

## Figures and Tables

**Figure 1 cancers-13-01823-f001:**
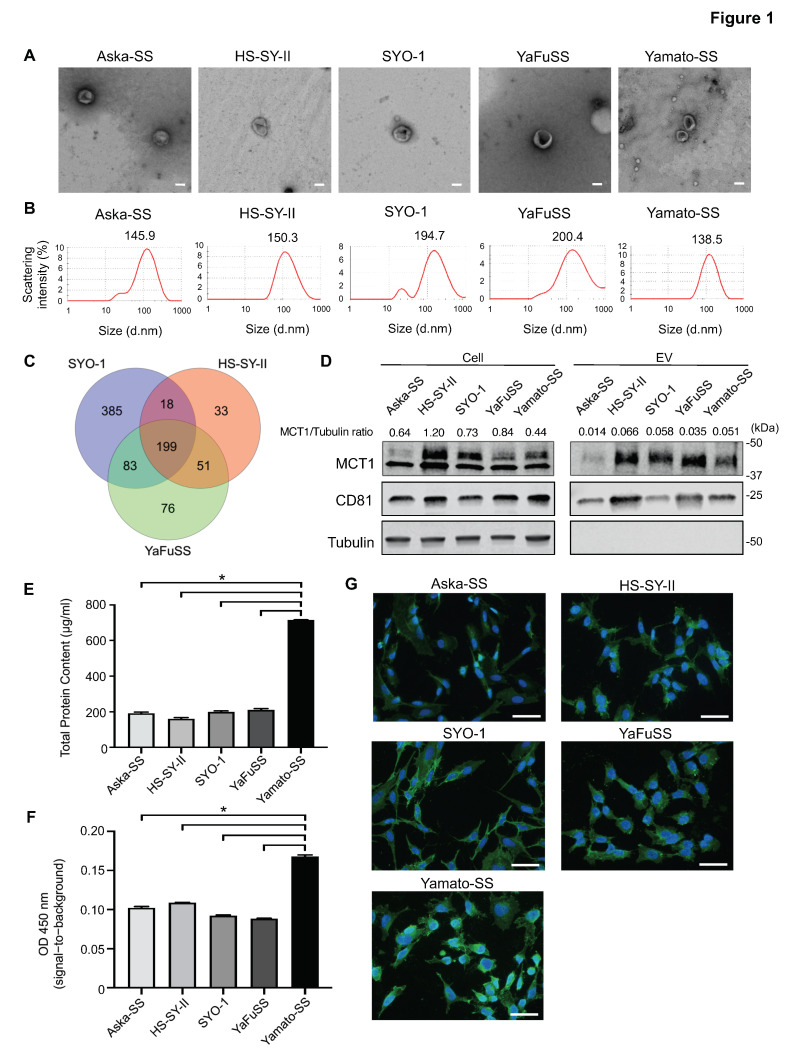
Identification of MCT1 as a surface marker of EVs secreted from synovial sarcoma (SS) cells. Data are presented as mean ± SD. (**A**) Transmission electron microscopy images of EVs secreted from SS cells with a cup-shaped morphology. Scale bars: 100 nm. (**B**) Particle diameter distribution analysis of EVs secreted by SS cells. (**C**) Venn diagram showing proteomics analysis of EVs from HS-SY-II, SYO-1, and YaFuSS cell lines. A total of 199 proteins were detected in EVs from these cell lines. (**D**). Western blot of MCT1 and CD81 in SS cell lysates and EVs from Aska-SS, HS-SY-II, SYO-1, YaFuSS, and Yamato-SS cell lines. Uncropped Western Blots are available in Appendix A. (**E**) Total protein content in EVs assessed using a Bradford assay (*n* = 3 in each group). * *p* < 0.05, one-way ANOVA. (**F**) Relative expressions of MCT1^+^CD81^+^ EVs in an ELISA (*n* = 3 in each group). * *p* < 0.05, one-way ANOVA. (**G**) Immunofluorescent staining for MCT1 (green) in Aska-SS, HS-SY-II, SYO-1, YaFuSS, and Yamato-SS cell lines. Nuclei are stained with DAPI (blue). Scale bar, 50 μm.

**Figure 2 cancers-13-01823-f002:**
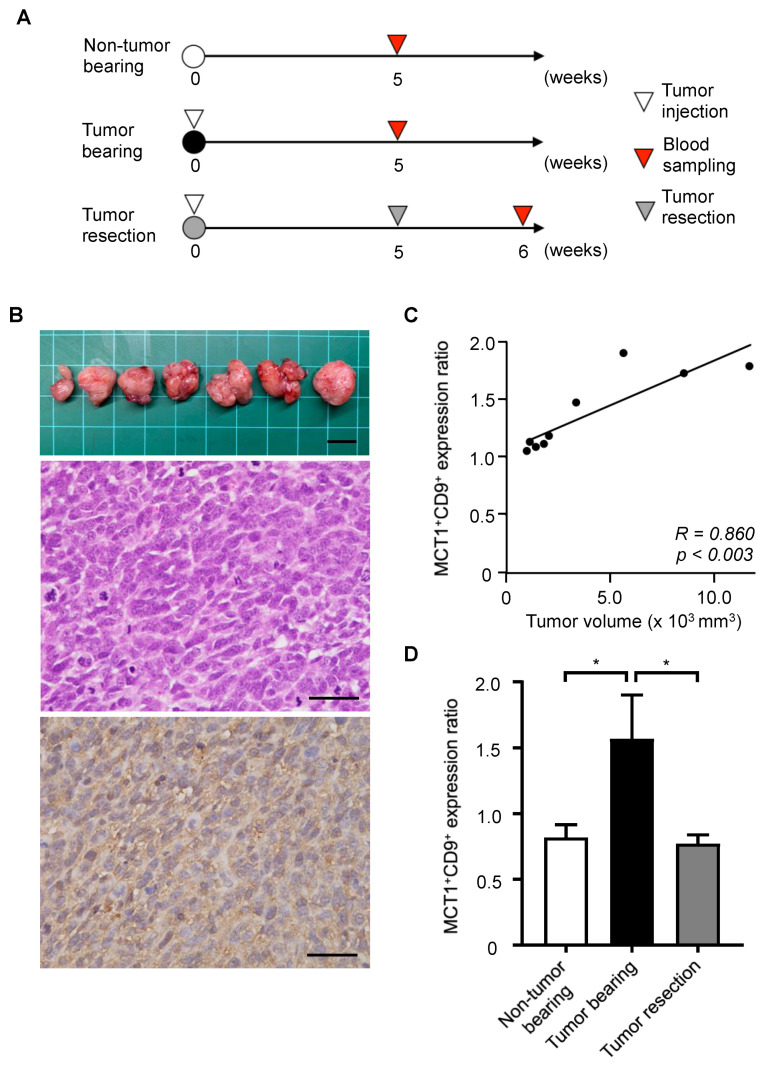
In vivo dynamics of MCT1^+^CD9^+^ EVs in synovial sarcoma-bearing mice. (**A**) Schema of the animal experiments. (**B**) The resected tumor (top), hematoxylin-eosin staining (center), and MCT1 immunohistochemistry for MCT1 (bottom). Scale bar: 10 mm (top) and 50 μm (center and bottom). (**C**) Pearson correlation between MCT1^+^CD9^+^ expression levels and tumor volume (*R* = 0.860, *p* = 0.003). (**D**) MCT1^+^CD9^+^ levels at the indicated time points. Data are presented as mean ± SD (n = 9 in each group). * *p* < 0.05, by one-way ANOVA.

**Figure 3 cancers-13-01823-f003:**
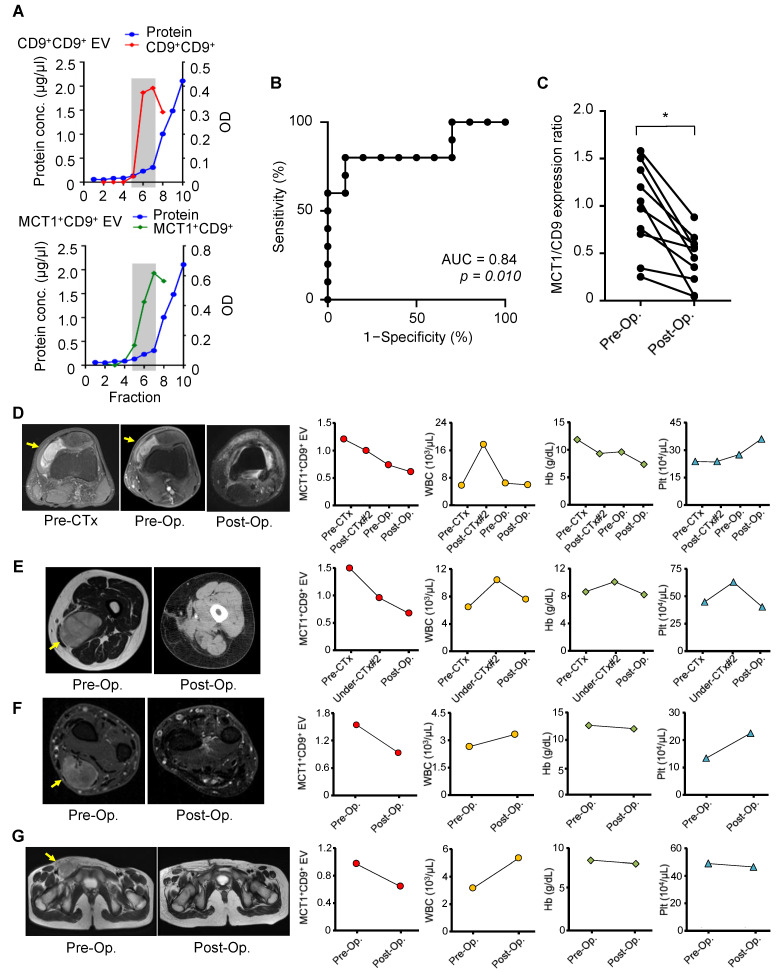
Tumor monitoring using MCT1^+^CD9^+^ EVs in patients with synovial sarcoma patients. (**A**) Protein concentration of purified EVs from human serum measured with a Bradford assay (blue). The red (CD9^+^CD9^+^) and green lines (MCT1^+^CD9^+^) indicate the optical density (OD) of the CD9^+^CD9^+^ and MCT1^+^CD9^+^ EVs using a sandwich ELISA, respectively. High-purity exosomes are collected in fractions 5–7. (**B**) Receiver operating characteristic (ROC) curve analysis. ROC curve analysis indicated an AUC of 0.84 (95% confidence interval: 0.65–1) to discriminate SS from pre- and post-operative states. (**C**) Serum EV MCT1^+^CD9^+^ expression ratio in pre- and post-operative states. * *p* < 0.05; Wilcox signed-rank test. (**D**–**G**) Tumor monitoring of the serum MCT1^+^CD9^+^ ratio during multimodal therapies. Four SS patients, including a 21-year-old male with left knee involvement (**D**), a 44-year-old female (left thigh) (**E**), a 64-year-old male (right forearm) (**F**), and an 11-year-old female (groin region) (**G**), were evaluated during the treatment. WBC: White blood cell, Hb: Hemoglobin, Plt: Platelet, Op: Operative surgery, CTx: Chemotherapy.

**Figure 4 cancers-13-01823-f004:**
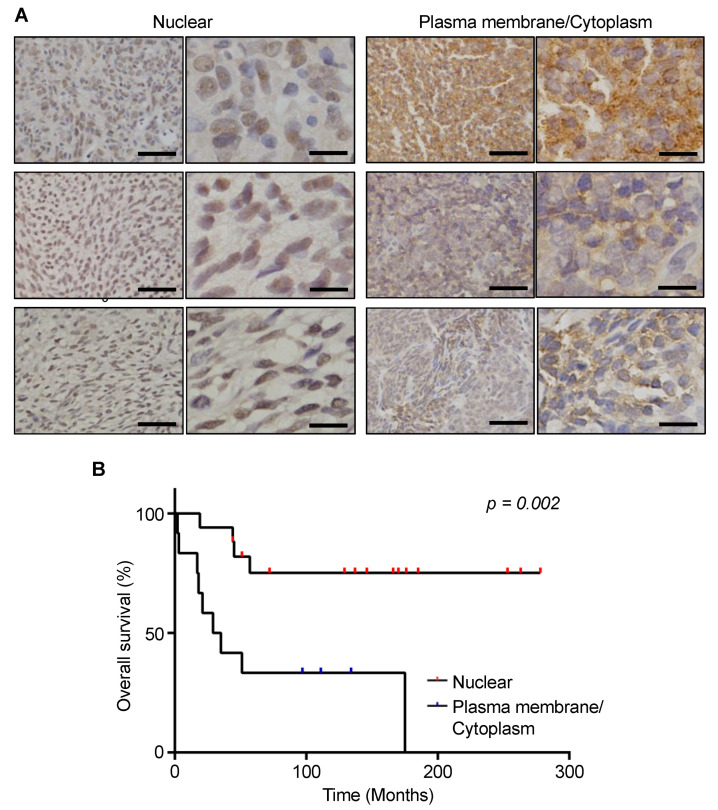
Immunohistochemical staining of MCT1 in SS tissue specimens. (**A**) SS tissues showing MCT1 expression in the nucleus (**left**) and cytoplasm/membrane (**right**). Scale bar: 50 μm (**left**), 15 μm (**right**). (**B**) Kaplan–Meier curves showing the overall survival stratified by the localization of MCT1 in the SS cells. Cytoplasmic expression of MCT1 was significantly associated with poorer overall survival, while nuclear expression of MCT1 was significantly associated with better overall survival (*p* = 0.002; log-rank test).

**Figure 5 cancers-13-01823-f005:**
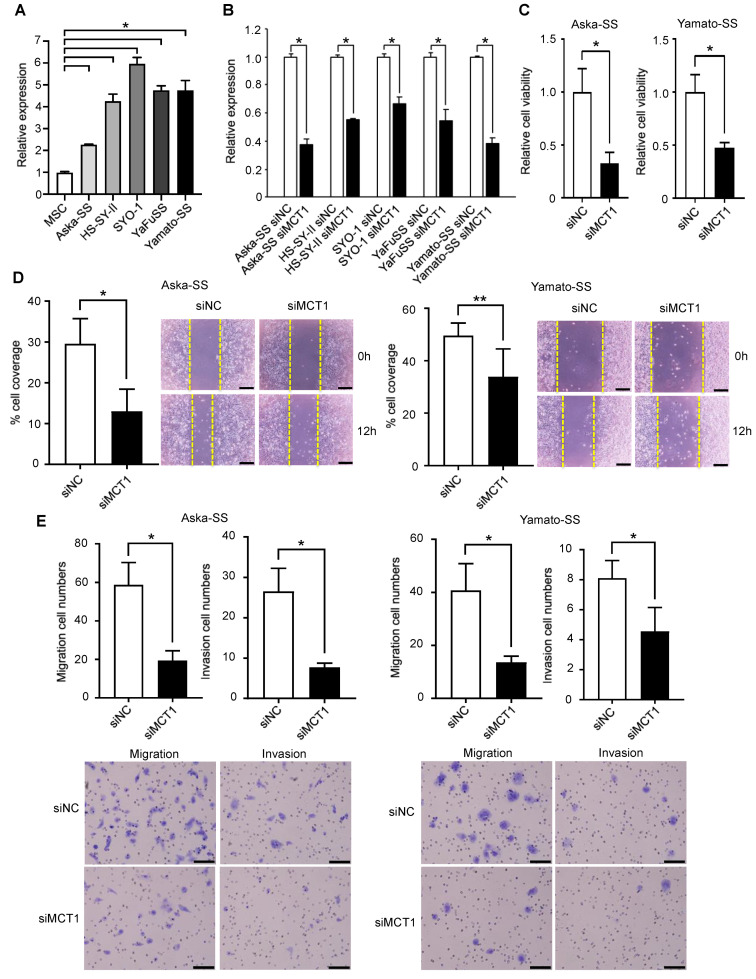
Silencing MCT1 in SS cells. (**A**) MCT1 mRNA expression levels in SS cells (Aska-SS, HS-SY-II, SYO-1, YaFuSS, and Yamato-SS) and hMSCs. hMSCs were used as a negative control. Data are presented as mean ± SD. * *p* < 0.01, by one-way ANOVA. (**B**) MCT1 expression levels in SS cells after transfection with siMCT1 and siNC. (**C**) Survival of Aska-SS and Yamato-SS cells after MCT1 knockdown. The number of adherent cells 48 h after transfection with siRNA is shown. * *p* < 0.001, by Student *t*-test. (**D**) Wound-healing assay. Silencing of MCT1 reduced the migration ability of the cells as determined with a wound-healing assay. The yellow dashed lines denote the margins of the wound. * *p* < 0.01 and ** *p* < 0.05 by Student *t*-test. (**E**) Migration and invasion analysis of SS cells. Silencing of MCT1 reduced both invasion and migration. * *p* < 0.01, by Student *t*-test. Scale bar: 100 μm.

**Table 1 cancers-13-01823-t001:** Characteristics of patients with SS whose tumor burden was monitored by MCT1^+^CD9^+^ EVs.

No.	Age (years)	Sex	Site	Size (cm)	Subtype	Fusion Gene	CTx	RTx	Viable Cells (%)	Local Recurrence	Distant Metastasis	Follow-Up Period (Months)	Oncologic Outcome
1	21	M	Knee	4.5	Mono-phasic	*SY18-SSX1*	Yes	No	95	Absence	−	47	CDF
2	44	F	Thigh	9.4	Mono-phasic	*SY18-SSX1*	Yes	No	50	Absence	+	23	DOD
3	64	M	Forearm	5.7	Mono-phasic	Unknown	No	No	-	Absence	−	53	CDF
4	48	M	Thigh	14.8	Bi-phasic	*SY18-SSX1*	Yes	No	95	Absence	−	55	CDF
5	11	M	Lumbar	4.3	Mono-phasic	*SY18-SSX1*	Yes	No	95	Absence	−	54	CDF
6	18	M	Thigh	13.7	Mono-phasic	Unknown	Yes	No	95	Absence	−	49	CDF
7	61	F	Thigh	6.6	Bi-phasic	Unknown	Yes	No	-	Absence	+	40	AWD
8	20	F	Back	8.0	Mono-phasic	Unknown	Yes	No	75	Absence	−	35	CDF
9	55	M	Abdomen	4.2	Mono-phasic	*SY18-SSX1*	No	No	-	Absence	−	11	CDF
10	11	F	Groin	9.5	Mono-phasic	*SY18-SSX2*	Yes	No	100	Absence	−	9	CDF

Abbreviations: CTx, chemotherapy; RTx, radiation therapy; CDF, continuous disease free; AWD, alive with disease; DOD, dead of disease.

**Table 2 cancers-13-01823-t002:** Clinicopathological correlation of the MCT1 expression between nuclear and cytoplasm/plasma membrane based on the univariate analysis in the validation cohort.

Variables	Nuclear (*N*)	Cytoplasm/Membrane (*N*)	*p*-Value
Age at diagnosis	-	-	0.053
0–40 years	7	10	-
41+ years	10	2	-
Sex	-	-	0.264
Male	7	8	-
Female	10	4	-
Subtype	-	-	0.439
Monophasic	9	8	-
Biphasic	6	4	-
Unknown	2	0	-
Size	-	-	0.519
≤5cm	3	3	-
>5cm	13	7	-
Unknown	1	2	-
Site	-	-	0.566
Thigh	9	8	-
Lower leg	1	0	-
Upper arm	0	1	-
Forearm	3	1	-
Others	4	2	-
Fusion gene	-	-	0.003
*SS18-SSX1*	14	2	-
*SS18-SSX2*	3	8	-
Unknown	0	2	-
Chemotherapy	-	-	0.218
Yes	14	7	-
No	3	5	-
Radiation therapy	-	-	0.370
Yes	4	1	-
No	13	11	-
Local recurrence	-	-	0.498
Presence	2	0	-
Absence	15	12	-
Distant metastasis	-	-	0.025
Presence	5	9	-
Absence	12	3	-
Oncologic outcome	-	-	0.010
DOD	4	9	-
CDF/NED	13	3	-

Abbreviations: DOD, dead of disease; CDF, continuous disease free; NED, no evidence of disease.

**Table 3 cancers-13-01823-t003:** Univariate and multivariate analysis predicting overall survival according to the clinicopathological factors. Univariate analysis; Kaplan-Meier method and log-rank test. Multivariate analysis; Cox regression hazard model.

Variables	*N*	Univariate		Multivariate		
		5-Year OS	*p*-Value	HR	95% CI	*p*-Value
Age at diagnosis	-	-	0.986	-	-	-
0–40 years	17	58%	-	-	-	-
41+ years	12	58%	-	-	-	-
Sex	-	-	0.041	-	-	-
Female	14	70%	-	1	Reference	-
Male	15	47%	-	2.49	0.74–8.39	0.141
Subtype	-	-	0.131	-	-	-
Monophasic	17	40%	-	-	-	-
Biphasic	10	80%	-	-	-	-
Unknown	2	100%	-	-	-	-
Size	-	-	0.266	-	-	-
≤5 cm	6	83%	-	-	-	-
>5 cm	20	53%	-	-	-	-
Unknown	3	33%	-	-	-	-
Site	-	-	0.735	-	-	-
Lower extremity	18	60%	-	-	-	-
Upper extremity	5	40%	-	-	-	-
Others	6	67%	-	-	-	-
Fusion gene	-	-	0.408	-	-	-
*SS18-SSX1*	16	68%	-	-	-	-
*SS18-SSX2*	11	44%	-	-	-	-
Unknown	2	50%	-	-	-	-
Chemotherapy	-	-	0.005	-	-	-
Yes	21	80%	-	1	Reference	-
No	8	25%	-	7.99	1.96–32.54	0.004
Radiation therapy	-	-	0.821	-	-	-
Yes	5	40%	-	-	-	-
No	24	62%	-	-	-	-
Local recurrence	-	-	0.240	-	-	-
Absence	27	55%	-	-	-	-
Presence	2	100%	-	-	-	-
Distant metastasis	-	-	0.010	-	-	-
Absence	15	79%	-	1	Reference	-
Presence	14	36%	-	2.94	0.71–12.22	0.138
MCT1 expression	-	-	0.002	-	-	-
Nuclear	17	75%	-	1	Reference	-
Cytoplasm/membrane	12	33%	-	5.34	1.20–23.89	0.028

## Data Availability

The data presented in this study are available on request from the corresponding author. The data are not publicly available due to ethical reasons.

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
