# Peer review of "Liquid Biopsy Targeting Monocarboxylate Transporter 1 on the Surface Membrane of Tumor-Derived Extracellular Vesicles from Synovial Sarcoma"

_cancers, 2021, doi:10.3390/cancers13081823_

Round 1
Reviewer 1 Report
In the present original work, the Authors show, for the first time, that circulating Synovial Sarcoma-derived EVs may represent a novel target for liquid biopsy of this aggressive disease, identifying MTC-1 as a powerful molecular marker to monitor also the progression of the disease.
In conclusion, the study is correctly designed and the results are well interpreted, and data are analyzed in appropriate way. The Author’s discovery opens new possibilities for the early diagnosis and more tailored prognosis/therapy for Synovial Sarcoma. Therefore, I recommend it for publication after minor changes:
- Page 3, Lane 123: Complete characterization of cell derived-EVs has been performed, while for patient derived-EVs it is just mentioned a WB control for CD9, which is eventually not shown in the paper figures. Please include the quality check WB images in the paper.
- Page 4, Lane 170: Please check the third word of the lane, I believe it has been automatically changed from “resection” to “rejection”.
- About Fig.2, the protocol for animal experiments, I believe that it would be interesting to get the information about MCT1+/CD9+ -EVs at the exact same time of the “tumor resection” samples. In other words, I would have performed blood sampling not only after 5 weeks, but also at 5+1 weeks. Could this information kindly be provided by the Authors?
- In the Materials and Methods section are missing some protocols. Please integrate experimental methods for siRNA transfection detailed protocol, qPCR protocol for mRNA level detection, and Wound Healing assay (see Fig.5).
- Page 7, Lane 299: In panel E description, I believe the protein content of EVs has been defined by Bradford assay and not by the Proliferation assay WST-1. If this is the case, please correct.
- Page 8, Lane 305: In the brackets the panel descriptions are “left” and “right”, while now the panel layout suggests to use “top” and “bottom”. Please correct the information.
- Page 8, Lane 30: Please specify that in point (D) the data are related to EVs.
- Page 9, Lane 337: Please include abbreviations for CTx (ChemoTherapy) and RTx (RadioTherapy).
- Fig. 3, panel A: Please describe more extensively how the levels of CD9 or MCT1 have been normalized on CD9 levels. At which stage “CD9 normalizer” levels have been measured?
Author Response
Answers to Reviewer 1:
In the present original work, the Authors show, for the first time, that circulating Synovial Sarcoma-derived EVs may represent a novel target for liquid biopsy of this aggressive disease, identifying MTC-1 as a powerful molecular marker to monitor also the progression of the disease.
In conclusion, the study is correctly designed and the results are well interpreted, and data are analyzed in appropriate way. The Author’s discovery opens new possibilities for the early diagnosis and more tailored prognosis/therapy for Synovial Sarcoma. Therefore, I recommend it for publication after minor changes:
- Page 3, Lane 123: Complete characterization of cell derived-EVs has been performed, while for patient derived-EVs it is just mentioned a WB control for CD9, which is eventually not shown in the paper figures. Please include the quality check WB images in the paper.
Answer: We appreciate your comment. In our previous study, patient-derived EVs were validated using TEM and Zetasizer nano ZSP (data not shown). One example in a 44-year-old female patient with synovial sarcoma is shown below (Figure A and F). Our previous data confirmed that this methodology is appropriate for obtaining EVs from patients’ sera. We have added these descriptions and a literature citation in the Methods section.
Figure A. EVs derived from serum of 44-year-old female patient with synovial sarcoma (shown in uploaded file).
Figure F. Western blotting of serum from a patient with SS for each fraction of the EV-second procedure. Fractions 3 to 7 were positive for CD9 (25 kDa). Strongly positive fractions 4 to 6 mainly contain exosomes (24) (shown in uploaded file).
Revised version:
2.5. Isolation of EVs from human serum
EVs were purified from human serum samples using size exclusion chromatography on a drip with Extracellular Vesicle isolation by Size Exclusion Chromatography ON Drip column EVSecond L70 (GL Sciences, Tokyo, Japan). The column was initially equilibrated with 700 μl of FBS twice, followed by three washing steps using 1,500 μl of PBS. After washing, 100 μl of the collected human serum sample was loaded onto the column, followed by the collection of 12 consecutive fractions in 100 μl of PBS. The CD9 expression in these fractions was analyzed using Western blotting, and CD9-positive fractions were recognized as the exosome-rich portion The obtained EVs using this system were previously authenticated with TEM and Western blotting, and the diameters of the particles in the EV fractions were analyzed with Zetasizer nano ZSP (24).
- Page 4, Lane 170: Please check the third word of the lane, I believe it has been automatically changed from “resection” to “rejection”.
Answer: We appreciate the comment. We have changed the word from “rejection” to “resection.”
Revised version:
2.9. Animal experiments
One of three groups (n = 9; tumor bearing/resection group) had tumor resection 5 weeks after tumor inoculation, followed by blood collection 1 week after tumor resection to avoid the bias by the post-surgical inflammation.
- About Fig.2, the protocol for animal experiments, I believe that it would be interesting to get the information about MCT1+/CD9+-EVs at the exact same time of the “tumor resection” samples. In other words, I would have performed blood sampling not only after 5 weeks, but also at 5+1 weeks. Could this information kindly be provided by the Authors?
Answer: We appreciate the reviewer’s suggestion. It should be interesting to investigate the amount of MCT1+/CD9+ EVs not only after 5 weeks (at the timing of tumor resection) but also after 6 weeks. However, we supposed that the results at 5 weeks (at the timing of tumor resection) might be influenced by the surgical procedure or procedure-related inflammation. Thus, we collected the blood samples 1 week after tumor resections. We have added these descriptions in the Methods section.
Revised version:
Methods 2.9. Animal experiments
One of three groups (n = 9; tumor bearing/resection group) had tumor resection 5 weeks after tumor inoculation, followed by blood collection 1 week after tumor resection to avoid the bias by the post-surgical inflammation.
- In the Materials and Methods section are missing some protocols. Please integrate experimental methods for siRNA transfection detailed protocol, qPCR protocol for mRNA level detection, and Wound Healing assay (see Fig.5).
Answer: We appreciate the comment. Accordingly, we have added the protocols and integrated them as follows:
Revised version:
2.13. RNA extraction and RT-qPCR
Total RNA was extracted using an miRNeasy mini kit (Qiagen, Valencia, CA, USA) in accordance with the manufacturer’s instructions. RNA samples were reverse transcribed using the TaqMan MicroRNA Reverse Transcription Kit (Applied Biosystems, Foster, CA, USA). For mRNA detection, total RNA was reverse transcribed using a PrimeScript RT Reagent Kit (Takara, Tokyo, Japan) in accordance with the manufacturer’s protocol. Quantitative polymerase chain reaction (PCR) was performed on an Agilent Mx3000P QPCR System (Agilent Technologies, Santa Clara, CA, USA) using the TaqMan 2×Universal PCR Master mix and each primer. Data obtained from RT-PCR were analyzed using the 2−∆∆Ct method. The mRNA expression levels were normalized using GAPDH.
2.14. siRNA transfection
Cells were seeded into 6- or 24-well plates on the day before the experiments. Synthetic siRNAs for MCT1 (si-MCT1, Thermo Fisher Scientific) were added to the cells at a final concentration of 20 nM in complete media the following day. DharmaFECT1 (GE Healthcare) were used for the transfection.
2.14.1. Cell proliferation assay
Cellular proliferation was measured using the WST-1 Proliferation Assay (Sigma-Aldrich) in accordance with the manufacturer’s instructions. Cells were seeded in 96-well plates after 24 h of transfection. Absorbance was measured at 450 nm, with a reference wavelength of 630 nm, using a microplate reader (BIO-RAD). The relative number of viable cells is expressed as the percentage of viable cells.
2.14.2. Wound-Healing assay
A scratch wound-healing assay was used to examine the cellular mobility characteristics. Cells were seeded in 24-well plates after 24 h of transfection. The medium was replaced with serum-free DMEM. Cell monolayers were scratched (wounded) using a sterile 200-μL pipette tip, and PBS was used for washing and removing cell debris. After 16 h, migrating cells were monitored and photographed under phase-contrast microscopy. The Image J software was used to quantify the relative wound size. Cell mobility inhibition (%) was calculated as the new scratch width/original scratch width × 100%. Experiments were repeated three times.
2.14.3. Transwell assay
Cell invasion and migration were examined using 24-well BD BioCoat invasion chambers with and without a Matrigel matrix (BD Biosciences, San Jose, CA, USA). SS cells (Aska-SS and Yamato-SS) were trypsinized and seeded (5 × 104 cells) in chambers 48 h after siRNA transfection. The cells were subsequently suspended in FBS-free medium and added onto the upper chamber, while medium containing 10% FBS was placed in the lower chamber. After 36 h of incubation, the cells on the upper chamber were removed, and the filters were fixed in methanol and stained with Hemacolor solution 3 (Merck, Darmstadt, Germany). The number of cells was counted in six separate high-power fields.
- Page 7, Lane 299: In panel E description, I believe the protein content of EVs has been defined by Bradford assay and not by the Proliferation assay WST-1. If this is the case, please correct.
Answer: We apologize for our mistake. We have corrected “WST-1” to “Bradford assay” in the legend of Figure 1E.
Revised version:
Figure 1. (E) Total protein content in EVs assessed using a Bradford assay (n = 3 in each group). *p < 0.05, by one-way ANOVA.
- Page 8, Lane 305: In the brackets the panel descriptions are “left” and “right”, while now the panel layout suggests to use “top” and “bottom”. Please correct the information.
Answer: We appreciate the comment. Accordingly, we have corrected the description in the legend of Figure 2B.
Revised version:
Figure 2. (B) The resected tumor (top), hematoxylin-eosin staining (center), and MCT1 immunohistochemistry for MCT1 (bottom). Scale bar: 10 mm (top) and 50 μm (center and bottom).
7. Page 8, Lane 30: Please specify that in point (D) the data are related to EVs.
Answer: We appreciate the comment. Figure 2D shows the results of the tumor monitoring using serum MCT1+CD9+ EVs. We have changed “CD9/CD9” to “CD9+CD9+ EVs,” and “MCT1/CD9” to “MCT1+CD9+ EVs” in the main text to describe that these are relevant to EVs.
Revised version:
Figure 2 (C) Pearson correlation between MCT1+CD9+ expression level and tumor volume (R = 0.860, p = 0.003). (D) MCT1+CD9+ levels at the indicated time points. Data are presented as mean ± SD (n = 9 in each group). *p < 0.05, by one-way ANOVA.
- Page 9, Lane 337: Please include abbreviations for CTx (ChemoTherapy) and RTx (RadioTherapy).
Answer: We appreciate the comment. Accordingly, we have included the definitions of the abbreviations CTx and RTx in Table 1.
Revised version:
Abbreviations: CTx, chemotherapy; RTx, radiation therapy; CDF, continuous disease-free; AWD, alive with disease; DOD, dead of disease
- 3, panel A: Please describe more extensively how the levels of CD9 or MCT1 have been normalized on CD9 levels. At which stage “CD9 normalizer” levels have been measured?
Answer: We appreciate the comment. In this study, we did not use CD9 as a normalizer, and these (MCT1+CD9+) were determined by a combination of the results from the Bradford assay (CD9) and sandwich ELISA (MCT1) using EVs extracted from human serum. In Figure 3A, the blue lines indicate the protein concentrate measured using the Bradford assay; and the red (CD9+CD9+) and green lines (MCT1+CD9+) lines, the optical density using a sandwich ELISA. The legend of Figure 3A has been modified.
Revised version:
Figure 3. Tumor monitoring using MCT1+CD9+ EVs in patients with synovial sarcoma. (A) Protein concentration of purified EVs from human serum measured with a Bradford assay (blue). The red (CD9+CD9+) and green lines (MCT1+CD9+) indicate the optical density (OD) of the CD9+CD9+ and MCT1+CD9+ EVs using a sandwich ELISA, respectively. High-purity exosomes are collected in fractions 5–7.

Reviewer 2 Report
This is very fine work. The details of the study are well presented. The only limitation is the study size. Otherwise I think this is an excellent study.
Author Response
Answers to Reviewer 2:
This is very fine work. The details of the study are well presented. The only limitation is the study size. Otherwise I think this is an excellent study.
Answer: We appreciate the comment. We agree that the small sample size is one of the limitations of this study. A validation study using a larger sample size would determine the clinical usefulness of liquid biopsy targeting MCT1+CD9+ EVs. We have added the description in the Discussion section as follows:
Revised version:
Discussion
In spite of the small sample size, we found that the circulating MCT1+CD9+ EVs in the serum accurately reflected the tumor burden or treatment response in vivo, indicating the potential for the translation of this liquid biopsy into clinics for the management of SS.

Reviewer 3 Report
This is a very strong extensive work on MCT1 expression in Synovial Sarcoma using EV liquid biopsy inovative technology.
While the article goes from cell lines, mice, patients to siRNA, it is extremely comprehensive on a methodological point of view.
I only have minor comments:
- The supplementary data mentioned in "3.2. Enrichment of MCT1 on EVs secreted from SS cell lines " paragraph that explains the choice of investingating MCT1 in "patients with SS compared to healthy individuals " might be included either in the introduction or in a prevous paragraph as it is relevant to understand why this target has been particularly chosen in this simple genetic sarcoma expected to be driven only by the pathognomonic translocation.
- p9. I would expect more prudence in concluding significance in such a small number of patients for the IHC MCT1 expression. What is missing is the pourcent of viable residual cells on tumor resection after neoadjuvant chemotherapy to better rely on the decrease of MCT1 expression.
Overall it is a very enthousiastic work that is Worth getting published.
Author Response
Answers to Reviewer 3:
This is a very strong extensive work on MCT1 expression in Synovial Sarcoma using EV liquid biopsy innovative technology.
While the article goes from cell lines, mice, patients to siRNA, it is extremely comprehensive on a methodological point of view.
I only have minor comments:
- The supplementary data mentioned in "3.2. Enrichment of MCT1 on EVs secreted from SS cell lines " paragraph that explains the choice of investigating MCT1 in "patients with SS compared to healthy individuals " might be included either in the introduction or in a previous paragraph as it is relevant to understand why this target has been particularly chosen in this simple genetic sarcoma expected to be driven only by the pathognomonic translocation.
Answer: We appreciate the comment. After the initial proteome analysis, we selected candidates by using the following algorithm: commonly expressed in EVs from all of the SS cell lines; plasma membrane protein expressed on the EV membrane; higher expression level in patients with SS than in healthy individuals; and decreased expression level postoperatively in patients with SS. Subsequently, MCT1 was a molecule that met all the criteria. However, the relationship between the expression of MCT1 and SS18-SSX has been unknown, which we aim to investigate as a next step. We have added these criteria in the Methods section as follows:
Revised version:
Methods
Candidate proteins were selected using the following criteria: commonly expressed in EVs from all of the SS cell lines; plasma membrane protein expressed on the EV membrane; higher expression level in patients with SS than in healthy individuals; and decreased expression level postoperatively in patients with SS.
- p9. I would expect more prudence in concluding significance in such a small number of patients for the IHC MCT1 expression. What is missing is the pourcent of viable residual cells on tumor resection after neoadjuvant chemotherapy to better rely on the decrease of MCT1 expression.
Answer: We appreciate the comment. We have added the percentage of viable cells in the resected tumor specimens that were treated with neoadjuvant chemotherapy in Table 1. In spite of the small number of patient samples, decreased MCT1+CD9+ EVs during neoadjuvant chemotherapy correlated with the percent of viable tumor cells. A slight decrease in MCT1+CD9+ EVs was observed in a 21-year-old man after neoadjuvant chemotherapy with three cycles of adriamycin and ifosfamide, whose resected tumor contained 95% viable cells (Figure 3D). On the other hand, a significant decrease in MCT1+CD9+ EVs was observed in a 44-year-old woman after neoadjuvant chemotherapy with two cycles of adriamycin and ifosfamide, whose resected tumor had 50% viable cells (Figure 3F).
Among 29 patients in whom the MCT1 expression was investigated using IHC (Table 2), 20 (69%) received chemotherapy. The rate of viable cells varied from 30% to 100% (mean, 69%), and none of the samples showed complete necrosis. MCT1 expression was appropriately evaluated on the live tumor cells.
We have added the percentages of viable cells in the Results section and revised Table 1 as follows:
Revised version:
Results
Case 1 (#1; Table 1) was a 26-year-old man with SS in his left knee (Fig. 3D). Serum MCT1+CD9+ EVs slightly decreased after neoadjuvant chemotherapy and further decreased after tumor resection. Microscopic evaluation revealed that the percentage of viable tumor cells was 95%. Case 2 (#2; Table 1) was a 47-year-old woman with SS in her left thigh (Fig. 3E). Similarly, serum MCT1+CD9+ EVs decreased after neoadjuvant chemotherapy and further decreased after tumor resection. The percentage of viable tumor cells was 50%.
